# Association of KMT2C Genetic Variants with the Clinicopathologic Development of Oral Cancer

**DOI:** 10.3390/ijerph19073974

**Published:** 2022-03-27

**Authors:** Mu-Kuei Shieu, Hsin-Yu Ho, Shu-Hui Lin, Yu-Sheng Lo, Chia-Chieh Lin, Yi-Ching Chuang, Ming-Ju Hsieh, Mu-Kuan Chen

**Affiliations:** 1Division of General Practice, Department of Medical Education, Changhua Christian Hospital, Changhua 500, Taiwan; prree2584@gmail.com or; 2Oral Cancer Research Center, Changhua Christian Hospital, Changhua 500, Taiwan; 183581@cch.org.tw (H.-Y.H.); 165304@cch.org.tw (Y.-S.L.); 181327@cch.org.tw (C.-C.L.); 177267@cch.org.tw (Y.-C.C.); 3Department of Surgical Pathology, Changhua Christian Hospital, Changhua 500, Taiwan; 74630@cch.org.tw; 4Department of Medical Laboratory Science and Biotechnology, Central Taiwan University of Science and Technology, Taichung 406, Taiwan; 5Department of Post-Baccalaureate Medicine, College of Medicine, National Chung Hsing University, Taichung 402, Taiwan; 6Graduate Institute of Biomedical Sciences, China Medical University, Taichung 404, Taiwan; 7Department of Otorhinolaryngology, Head and Neck Surgery, Changhua Christian Hospital, Changhua 500, Taiwan

**Keywords:** oral squamous cell carcinoma, lysine methyltransferase 2C, single-nucleotide polymorphisms, alcohol drinking

## Abstract

Lysine methyltransferase 2C (KMT2C) is a tumor-suppressor gene in several myeloid cells and epithelia and is linked with blood and solid tumor cancers. KMT2C single-nucleotide polymorphisms (SNPs) are also connected with several cancer types. Our study aimed to explore the potential genetic polymorphisms of KMT2C in oral cancer. Five KMT2C SNPs, including rs201834857, rs4725443, rs6464221, rs74483926, and rs6943984, were evaluated in 284 cancer-free controls and 284 oral squamous cell carcinoma (OSCC) cases. We found that individuals with the TC genotype or TC + CC genotype of rs4725443 had a higher risk of oral cancer incidence than those with the TT genotype. Further analysis of KMT2C SNP rs4725443 revealed that the TC + CC genotype of rs4725443 was associated with a significantly advanced tumor stage in the non-alcohol-drinking population. Moreover, the TC + CC genotype of rs4725443 was connected with poor cell differentiation in the alcohol-drinking population. Through analyzing a dataset from The Cancer Genome Atlas (TCGA), we found that reduced KMT2C levels were associated with advanced tumor stage, lymph node invasion, and poor cell differentiation in head and neck squamous cell carcinomas. Our data suggest that KMT2C SNP rs4725443 is a potential genetic marker for oral cancer patients in both non-alcohol-drinking and alcohol-drinking populations.

## 1. Introduction

Oral cancer is a life-threatening illness and a burden that affects those afflicted for decades [1]. It is estimated that oral diseases affect nearly 3.5 billion people worldwide, and a major proportion is male [1]. In Asian–Pacific countries such as Taiwan, the number of oral cancer patients is far above the global average [2,3]. According to the Health Promotion Administration (HPA) in Taiwan, the onset age of oral cancer is approximately 10 years earlier than other malignant cancers. In other words, early diagnosis plays an important role in oral cancer prevention. More than 90% of oral and pharyngeal cancers are squamous cell carcinomas and have a 5 year survival rate of approximately 55–60% [4]. TNM staging, on the other hand, directly results in the survival of oral cancer patients, according to the American Joint Committee on Cancer (AJCC) [5]. According to an analysis of the Surveillance, Epidemiology, and End Results (SEER) database, the oral tongue is the most common subsite and is more associated with higher mortality than others [6]. Oral cancer emerges due to multiple reasons. Individual habits such as tobacco, alcohol, and betel nut consumption, as well as smoking, are some of the prime triggers for oral cancer [7]. Furthermore, sexually transmitted diseases, such as HPV-mediated (p16) [8], are also associated with oral cancer. However, genome-wide and targeted-gene association studies [9,10] have found relationships between single-nucleotide polymorphisms and carcinogenesis.

Lysine Methyltransferase 2C (KMT2C), also known as mixed-lineage leukemia 3 (MLL3), is a tumor-suppressor gene in several myeloid cells and epithelia. It has a linkage with blood and solid tumor cancers (e.g., head, neck, esophagogastric, lung, endometrial, breast, bladder, and brain cancers) and is known for its negative effects on cell growth [11,12,13]. As one of the KMT2 family proteins with a large coding gene (1700 kb), KMT2C is located on chromosome 7q36 and encodes nuclear proteins including an AT hook DNA-binding domain, a DHHC-type zinc finger, six PHD-type zinc fingers, a SET domain, a post-SET domain, and a RING-type zinc finger [14,15,16]. In 2016, a study showed that KMT2C polymorphisms are associated with laryngeal cancer [17]. Additionally, a study conducted in 2019 identified that a KMT2C gene mutation was related to familial non-syndromic primary failure of tooth eruption [14]. Additional studies conducted in 2020 also showed that KMT2C inactivation may promote cancer development through transcriptional dysregulation in several pathways [15]. Yet, the mechanisms contributing to the development of oral cancer and single-nucleotide polymorphisms remain unclear. Our research established the KMT2C gene as haplotypes that focused on five SNPs (rs201834857, rs4725443, rs6464211, rs74483926, and rs6943984), and we attempted to assess the relationship between these SNPs and oral cancer in the Asian population.

## 2. Materials and Methods

### 2.1. Patients and Specimens

The 2013 to 2020 data of 284 oral-cancer-related cases and 284 cancer-free cases who participated in this study, approved by the Institutional Review Board of Changhua Christian Hospital (CCH), were extracted. Based on the TNM staging system of the American Joint Committee on Cancer (AJCC) criterion [18], clinical staging, lymphocyte metastasis, and tumor cell differentiation of OSCC were determined. For comparison, we selected 284 cases that had no self-reported history of cancer from Changhua Christian Hospital Biobank. Apart from age and sex, dichotomous outcomes such as alcohol drinking, betel nut consumption, and smoking were also recorded for every subject. Betel quid chewing and alcohol drinking were defined as ever chewing betel quid or drinking. Smoking was classified as more than one cigarette per day during the past 3 months. The study was approved by the Institutional Review Board (IRB) of Changhua Christian Hospital (Changhua, Taiwan; IRB No. 130616, date of approval 26 August 2021) and Changhua Christian Hospital Biobank (Changhua, Taiwan; IRB No. 200211, date of approval 13 March 2022).

### 2.2. Selection and Genotyping of KMT2C SNPs

As previously described [19], genomic DNA extraction was conducted. Five common polymorphisms (rs201834857, rs4725443, rs6464211, rs74483926, and rs6943984) from KMT2C gene potential were determined by real-time quantitative PCR using the ABI StepOne real-time PCR system (Applied Biosystems, Foster City, CA, USA) and analyzed using StepOne Software v2.3.

### 2.3. Bioinformatics Analysis

By using the University of California Santa Cruz (UCSC) Xena Functional Genomics Explorer [20], we analyzed the correlation between KMT2C expression and clinical presentations of head and neck squamous cell carcinomas (HNSCCs) in a dataset from The Cancer Genome Atlas (TCGA).

### 2.4. Statistical Analysis

Data were calculated using IBM SPSS Statistics v22.0 (IBM, Armonk, NY, USA). The tumor stage, TNM status, and cell differentiation were expressed by descriptive analysis. We used the Mann–Whitney U test to evaluate significant variations in demographic data between OSCC cases and non-cancer controls. Additionally, we used logistic regression to calculate the odds ratio (OR) of the KMT2C SNP distribution, while multiple regression was used to yield adjusted odds ratios (AOR) with correlated 95% confidence intervals (CIs) logistic regression methods for measuring the KMT2C SNP distribution between OSCC cases and non-cancer controls, after adjusting for betel nut chewing, cigarette smoking, and alcohol drinking. The variants of KMT2C levels in HNSCC dataset from TCGA were compared with the Mann–Whitney U test. A *p*-value of < 0.05 was considered to indicate statistical significance.

## 3. Results

### 3.1. Cohort Characteristics

Our study consisted of 284 OSCC patients and 284 cancer-free patients among control cases. Among all cases, there was no statistical difference in age distribution between the control group and patients with oral cancer. According to the seventh edition of the AJCC and TNM staging system [20], most of the patients were classified as having non-lymph-node metastasis (76.1%) and without distant organ metastasis (93.7%). Moreover, most of our OSCC cases had moderate to poor cell differentiation. Statistics were significant for betel nut chewing, cigarette smoking, and alcohol drinking between the control group and patients (Table 1).

### 3.2. Association of KMT2C Gene Polymorphism with the Progression of OSCC

Regarding genotype distribution, five SNPs (rs201834857, rs4725443, rs6464211, rs6943984, and rs74483926) of the KMT2C gene were evaluated one by one. ORs and their 95% CIs were estimated by logistic regression models. We added AORs as our secondary results, which combined different variables such as personal habits (alcohol drinking, betel nuts chewing, and cigarette smoking) into the calculation. Different genotype variables are listed in Table 2. We first revealed that the KMT2C SNPs rs4725443 and rs6943984 were statistically significant between the control group and patients with oral cancer by estimating the OR and AOR. In the rs4725443 subgroup, patients who carried at least one minor C allele of rs4725443 tended to have a higher oral cancer occurrence than the wild-type group with a major T allele proportion (TC versus TT; OR: 1.612; 95% CI, 1.111–2.338; *p* = 0.012) (TC + CC versus TT; OR: 1.610; 95% CI, 1.126–2.302; *p* = 0.009). In the rs6943984 subgroup, we found that persons who carried the AA allele of rs6943984 tended to have a lower oral cancer occurrence in comparison with the wild-type group with a major G allele proportion (AA versus GG; OR: 0.265; 95% CI, 0.073–0.964; *p* = 0.044). Meanwhile, the persons who carred the AA allele of rs6943984 showed similar results after adjusting for personal habits (AA versus GG; AOR: 0.046; 95% CI, 0.004–0.541; *p* = 0.014).

We then investigated the polymorphic correlations between genotypes and clinical pathological characteristics of OSCC patients, trying to figure out whether the genotypes of SNP rs4725443 had correlations with variable categories (Table 3). Nevertheless, the different distribution of the allele genotype did not influence the status of clinical stage, tumor size, lymph node invasion, or distal metastasis. Taking rs4725443 into further consideration, as alcohol drinking, betel nut chewing, and cigarette smoking are known as risk factors contributing to OSCC, we calculated patients’ different genotypes in relation to each other. Our study revealed that the group composed of cigarette-smoking and betel-nut-chewing subjects did not have correlations with the status of clinical stage, tumor size, lymph node metastasis, distant metastasis, or cell differentiation in the KMT2C SNP rs4725443 subgroup (Appendix A). However, the alcohol-drinking group showed that patients with the TT allele were prone to having poor cell differentiation compared with patients with the TC + CC allele, and the results were statistically significant (*p* = 0.028). In addition, we also revealed that in the group of the non-alcohol-drinking subjects, patients with TC and CC alleles were more prone to having a poor clinical stage than patients with TT alleles (*p* = 0.034) (Table 4). Further comparative results from the subgroup of patients with different habits demonstrated a correlation between KMT2C SNPs and environmental risks with the progression of oral cancer.

### 3.3. Clinical and Functional Insights from KMT2C to OSCC

As a genetic association between KMT2C and oral cancer was mentioned, TCGA datasets were used to examine the clinical outcomes of this gene (Figure 1). With the above charts, we found that levels of the KMT2C gene did not show differences in tumor size (*p* = 0.7396). On the other hand, the clinical stage (*p* = 0.0041), lymph node status (*p* = 0.0048), and cell differentiation between well and poor statuses (*p* = 0.036) and between moderate and poor statuses (*p* = 0.0301) were significantly different.

## 4. Discussion

The research of Folkman’s innovative tumor angiogenesis hypothesis was published in 1971 [21]; since then, there has been great interest in understanding the role of the KMT2C gene. In our study, we found that the single-nucleotide polymorphism rs4725443, located on the KMT2C gene, showed remarkable differences between oral cancer patients and the control group, whereby patients were prone to carrying the T allele. The results provide evidence that the KMT2C gene’s diversity influences the possibility of oral cancer occurrence.

Oral cancer is a multifactorial disease in which genetic background is significant, and its etiology has much space to develop [22,23]. Accumulative evidence has shown the correlation between oral cancer and its SNPs [24,25,26,27,28]. For instance, the proinflammatory cytokine interleukin 8 and its SNPs are associated with the susceptibility of oral cancer [27]. Regarding a highly polymorphic enzyme called N-acetyltransferase, variations in its SNPs are closely associated with oral cancer progression and development [28]. The location of a SNP within the gene’s coding sequence may lead to amino acid substitution and alter the protein’s function, and it plays a key role in the susceptibility to cancer [26]. In the previous investigation, the KMT2C gene was found to be linked with carcinogenesis progression of the lung, endometrial cancers, and medulloblastomas [29,30,31]. In contrast, to our knowledge, no research has reported on head and neck carcinomas. Poreba et al. [32] showed that the KMT2 family is involved in gene frameshifts, insertions, and deletions, which leads to the inactivation of proteins. The aim of our research was to select SNPs as genetic markers to predict the possibility of oral cancer.

In our genotype-classification study, persons carrying heterogeneous TC or TC + CC alleles instead of TT alleles had, respectively, a 1.612-fold (95% CI, 1.111–2.338) and 1.610-fold (95% CI, 1.126–2.302) higher incidence of oral cancer among KMT2C variant rs4725443. However, when we considered the OR that was adjusted for personal habits, there was no significant difference between the control and OSCC-patient groups, which indicated that the occurrence of oral cancer might be correlated with personal habits in terms of KMT2C variant rs4725443. Consistent with our observation, a previous study demonstrated that the minor C allele of KMT2C variant rs4725443 is associated with the risk of gastric cancer in the Han Chinese population [33]. Our statistics demonstrated that the major GG allele of KMT2C SNP rs6943984 was significantly associated with a higher risk of oral cancer than the AA allele (OR: 0.265; 95% CI, 0.073–0.964; *p* = 0.044). These results may be considered biased because of the small sample number of control (3.9%) and OSCC patients (1.1%) used. Increased control and patient numbers need to be included in further investigations to verify the meaning of KMT2C SNP rs6943984 in oral cancer. Thus, the following discussion focuses on KMT2C variant rs4725443.

Prior to our analysis of rs4725443, we knew that personal habits such as alcohol drinking, betel nut chewing, and smoking have a strong correlation with oral cancer occurrence [34,35,36,37]. However, when comparing different genotypes of alleles, only alcohol drinking was statistically significant among the different kinds of alleles in our study. These results may be partially explained by studies proving that the mechanism of alcohol drinking is modulated by gene polymorphisms due to ethanol metabolism [38,39]. Additionally, non-alcohol-drinkers are correlated with distributions of clinical staging. There are some rs4725443-related investigations included in cancer-related reports [17,33]. One of these reports published in China found that rs4725443 with the minor C allele exhibited a strong frequency (0.151 compared to 0.072, *p* < 0.001) and associations with laryngeal cancer cases, especially in Asian countries. For a total of 307 smoking patients and carriers of the TC + CC allele, their corresponding OR was 11.107 (95% CI, 7.007–17.607, *p* < 0.001), compared with 274 alcohol-drinking patients with the TC + CC allele, whose corresponding OR was 6.661 (95% CI, 2.965–14.965, *p* < 0.001) [17]. This difference demonstrated that the location of cancer might be correlated with different personal habits in terms of KMT2C variant rs4725443. Despite this, we conclude that the alcohol-drinking and non-alcohol-drinking patients may be associated with different clinicopathologic statuses related to rs4725443 allele phenotype presentation.

Our results also revealed an influence of the KMT2C gene on tumor stage, metastasis on the lymph node, and cell differentiation, which indicated the important role of KMT2C in oral cancer. Yet, there are some limitations to this study. First, a wide distribution of ages was not included in this study because the incidence of oral cancer is prone to occurring in middle age. Second, different levels of alcohol addiction or frequencies of other habits may have contributed to various results. Last but not least, further investigations concerning the effects of diversity among ethnicities and other potential risk factors such as the incidence rate of human papillomavirus or other metabolic syndromes resulting in oral carcinoma are required. Additionally, the early detection of risk factors and significant clinical findings (such as oral leukoplakia) are crucial for oral precancer progression. Our future study will focus on understanding the relationship between the KMT2C gene and oral precancer, which might demonstrate the role of KMT2C in the prevention of oral cancer.

## 5. Conclusions

In conclusion, based on our experimental results, we confirm that SNP rs4725443, which is located on the KMT2C gene, was strongly linked to OSCC occurrence and poor clinical stage in OSCC patients. People who had TC or TC + CC alleles of rs4725443 were correlated with an increased odds of OSCC occurrence. Moreover, OSCC patients with no alcohol-drinking habits who also had TC + CC alleles of rs4725443 had a high risk of poor tumor stage. The results of this study provide evidence of the effect of KMT2C SNPs and may be used to further investigate the application of the rs4725443 genetic marker for diagnosis and prevention.

## Figures and Tables

**Figure 1 ijerph-19-03974-f001:**
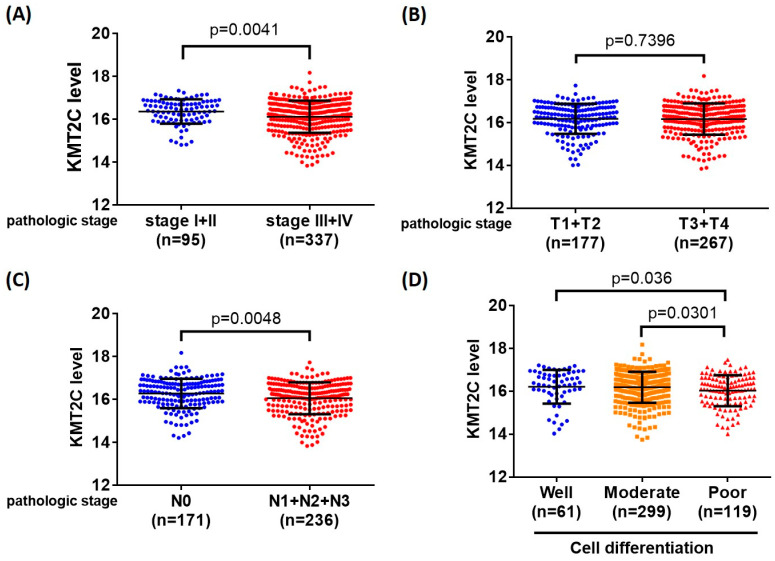
KMT2C expression levels were associated with clinicopathological parameters in HNSCC. The correlation between KMT2C expression and (**A**) clinical stage, (**B**) tumor size, (**C**) lymph node metastasis, and (**D**) cell differentiation of HNSCC in TCGA database. *p*-value < 0.05 is statistically significant.

**Table 1 ijerph-19-03974-t001:** The distributions of demographical characteristics and clinical parameters in 284 controls and 284 cases with OSCC.

Variable	Controls (*N* = 284)	Patients (*N* = 284)	*p*-Value
Age (years) Mean ± SD (Median)	53.74 ± 7.73 (53)	55.11 ± 10.30 (55)	*p* = 0.0742
Betel nut chewing			
No	273 (96.1%)	51 (18.0%)	*p* < 0.001 *
Yes	11 (3.9%)	233 (82%)	
Cigarette smoking			
No	262 (92.3%)	33 (11.6%)	*p* < 0.001 *
Yes	22 (7.7%)	251 (88.4)	
Alcohol drinking			
No	277 (97.5%)	139 (48.9)	*p* < 0.001 *
Yes	7 (2.5%)	145 (51.1)	
Stage			
I + II		168 (59.2)	
III + IV		116 (40.8)	
Tumor T status			
T1 + T2		193 (68.0%)	
T3 + T4		91 (32.0%)	
Lymph node status			
N0		216 (76.1%)	
N1 + N2 + N3		68 (23.9%)	
Metastasis			
M0		266 (93.7%)	
M1		18 (6.3%)	
Cell differentiation			
Well-differentiated		48 (16.9%)	
Moderately or poorly differentiated		236 (83.1)	

*N*: number. The Mann–Whitney U test was used between age of OSCC patients and cancer-free patients. The significance of betel nut chewing, cigarette smoking, and alcohol drinking between OSCC patients and cancer-free patients was calculated by logistic regression. * *p*-value < 0.05 is statistically significant.

**Table 2 ijerph-19-03974-t002:** The distribution of genotype frequencies in KMT2C SNPs in cases-of-OSCC group.

Variable	Controls (*N* = 284)	Patients (*N* = 284)	OR ^a^ (95% CI)	AOR ^b^ (95% CI)
rs201834857				
CC	134 (47.2%)	131 (46.1%)	1.000	1.000
CT	124 (43.7%)	124 (43.1%)	1.023 (0.723–1.446)	0.699 (0.355–1.376)
TT	26 (9.2%)	29 (10.2%)	1.141 (0.638–2.041)	0.811 (0.281–2.336)
CT + TT	150 (52.8)	153 (53.9%)	1.043 (0.750–1.451)	0.722 (0.382–1.362)
rs4725443				
TT	209 (73.6%)	180 (63.4%)	1.000	1.000
TC	67 (23.6%)	93 (32.7%)	1.612 (1.111–2.338) * *p* = 0.012	1.298 (0.660–2.552)
CC	8 (2.8%)	11 (3.9%)	1.597 (0.628–4.056)	0.841 (0.106–6.667)
TC + CC	75 (26.4%)	104 (36.6%)	1.610 (1.126–2.302) * *p* = 0.009	1.259 (0.651–2.435)
rs6464211				
CC	144 (50.7%)	128 (45.1%)	1.000	1.000
CT	112 (39.4%)	124 (43.7%)	1.246 (0.878–1.766)	1.485 (0.765–2.885)
TT	28 (9.9%)	32 (11.3%)	1.286 (0.734–2.252)	0.911 (0.302–2.753)
CT + TT	140 (49.3%)	156 (54.9%)	1.254 (0.901–1.743)	1.358 (0.721–2.557)
rs6943984				
GG	209 (73.6%)	215 (75.7)	1.000	1.000
GA	64 (22.5%)	66 (23.2%)	1.002 (0.677–1.485)	1.502 (0.729–3.095)
AA	11 (3.9%)	3 (1.1%)	0.265 (0.073–0.964) * *p* = 0.044	0.046 (0.004–0.541) * *p* = 0.014
GA + AA	75 (26.4%)	69 (24.3%)	0.894 (0.613–1.306)	1.215 (0.603–2.450)
rs74483926				
GG	189 (66.5%)	200 (70.4%)	1.000	1.000
GA	86 (30.3%)	77 (27.1%)	0.846 (0.587–1.220)	0.794 (0.388–1.624)
AA	9 (3.2%)	7 (2.5%)	0.735 (0.268–2.013)	0.429 (0.036–5.139)
GA + AA	95 (33.5%)	84 (29.6%)	0.836 (0.586–1.191)	0.764 (0.379–1.539)

*N*: number. * *p*-value < 0.05 is statistically significant. ^a^ The odds ratios (ORs) with their 95% confidence intervals were estimated by logistic regression models. ^b^ The adjusted odds ratios (AORs) with their 95% confidence intervals were estimated by multiple logistic regression models after controlling for betel nut chewing and alcohol and tobacco consumption.

**Table 3 ijerph-19-03974-t003:** Clinical statuses and KMT2C rs4725443 genotype frequencies in cases-of-OSCC group.

Variable	KMT2C (rs4725443)			
	TT (%) (*N* = 180)	TC + CC (%) (*N* = 104)	OR (95% CI)	*p*-Value
Clinical stage				
Stage I/II	111 (61.7%)	57 (54.8%)	1.000	*p* = 0.258
Stage III/IV	69 (38.3%)	47 (45.2%)	1.326 (0.813–2.164)	
Tumor size				
T1 + T2	126 (70.0%)	67 (64.4%)	1.000	*p* = 0.332
T3 + T4	54 (30.0%)	37 (35.6%)	1.289 (0.772–2.152)	
Lymph node metastasis				
No	139 (77.2%)	77 (74.0%)	1.000	*p* = 0.545
Yes	41 (22.8%)	27 (26.0%)	1.189 (0.679–2.081)	
Distant metastasis				
No	169 (93.9%)	97 (93.3%)	1.000	*p* = 0.836
Yes	11 (6.1%)	7 (6.7%)	1.109 (0.416–2.954)	
Cell differentiation				
Good	28 (15.6%)	20 (19.2%)	1.000	*p* = 0.427
Moderate/poor	152 (84.4%)	84 (80.8%)	0.774 (0.411–1.457)	

*N*: number. The odds ratios (ORs) with their 95% confidence intervals were estimated by logistic regression models.

**Table 4 ijerph-19-03974-t004:** Clinical statuses and KMT2C rs4725443 genotype frequencies in cases-of-OSCC group among 139 non-alcohol-drinking and 145 alcohol-drinking cases.

Variable	KMT2C (rs4725443)					
	Non-Alcohol-Drinking (*N* = 139)			Alcohol Drinking (*N* = 145)		
	TT (%) (*N* = 88)	TC + CC (%) (*N* = 51)	*p*-Value	TT (%) (*N* = 92)	TC + CC (%) (*N* = 53)	*p*-Value
Clinical stage						
Stage I/II	64 (72.7%)	28 (54.9%)	*p* = 0.034 *^,a^	47 (51.1%)	29 (54.7%)	*p* = 0.673
Stage III/IV	24 (27.3%)	23 (45.1%)		45 (48.9%)	24 (45.3%)	
Tumor size						
T1 + T2	70 (79.5%)	33 (64.7%)	*p* = 0.057	56 (60.9%)	34 (64.2%)	*p* = 0.695
T3 + T4	18 (20.5%)	18 (35.3%)		36 (39.1%)	19 (35.8%)	
Lymph node metastasis						
No	74 (84.1%)	39 (76.5%)	*p* = 0.269	65 (70.7%)	38 (71.7%)	*p* = 0.894
Yes	14 (15.9%)	12 (23.5%)		27 (29.3%)	15 (28.3%)	
Distant metastasis						
No	84 (95.5%)	48 (94.1%)	*p* = 0.729	85 (92.4%)	49 (92.5%)	*p* = 0.989
Yes	4 (4.5%)	3 (5.9%)		7 (7.6%)	4 (7.5%)	
Cell differentiation						
Good	22 (25.0%)	10 (19.6%)	*p* = 0.468	6 (6.5%)	10 (18.9%)	*p* = 0.028 *^,b^
Moderate/poor	66 (75.0%)	41 (80.4%)		86 (93.5%)	43 (81.1%)	

*N*: number. Values were estimated by logistic regression models. * *p*-value < 0.05 is statistically significant. ^a^ OR (95% CI): 2.190 (1.062–4.519); ^b^ OR (95% CI): 0.300 (0.102–0.880).

## Data Availability

The datasets generated for this study are available upon request to the corresponding authors.

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
