# Peer review of "Association of KMT2C Genetic Variants with the Clinicopathologic Development of Oral Cancer"

_ijerph, 2022, doi:10.3390/ijerph19073974_

Round 1
Reviewer 1 Report
Thank you for inviting me to review the manuscript entitled “Association of KMT2C genetic variants with clinicopathologic development of oral cancer”. In this manuscript, the authors investigated the significance of KMT2C genetic variants in the progression of development of oral cancer.”
- In this paper, 284 cancer-free patients were recruited as control group. The criteria for recruiting 284 adults need to be clearly stated.
- It is necessary to clearly describe how Betel quid Chewing's standards were set. Although the manuscript says "excessive use of betel quid," it is necessary to explain the criteria that the authors determine as "excessive use."
Reviewer 2 Report
The authors have investigated KMT2C genetic variants with clinicopathologic 2
development of oral cancer. The study is well designed. However presentation of manuscript can be improved.
Minor comments
- Over all , few grammatical errors could be identified. Some sentences do not have any convey what they have to convey
- Discussion seems a bit weak, The interactions of this gene with other tumor genes can be incorporated.
- Was the gene studied in oral precancers? Implications can be discussed
- Future perspectives and recommendations for further research needs to be included as well.
Reviewer 3 Report
The manuscript addresses an interesting topic which falls within the scope of IJERPH. The objective of this study was to explore the potential genetic polymorphism of KMT2C in oral cancer. The authors have highlighted the aims, significance and the novelty of their work. However, there are some important issues that must be addressed:
The authors have not properly presented why they chose some specific statistical tests, and how the normality of data distribution was assessed.
Numerous data presented in the text in the Results section, especially in the Cohort Characteristics subsection, is redundant and can also be found in the Tables.
The whole first paragraph in the Discussion section regards other types of cancer and would fit better in the Introduction section.
Moreover, improvements of the language and editing are necessary.
Some irrelevant, incorrect or colloquial expressions are used. For example:
“Oral cancer has become a life-threatening illness and an inevitable problem”
“In other words, early detective medical examination can play an important role on oral cancer prevention”
“Squamous cell carcinomas composed of 90% of these oral and pharyngeal cancers”
Other phrases are unclear or difficult to understand. For example:
“In Taiwan, the average incidence of oral cancer was approximately 10 years earlier than other malignant cancer”
“There are 284 oral cancer-related cases with 284 cancer-free patients participated in this study”
Round 2
Reviewer 3 Report
The authors have addressed some of my previous comments.
However, the manuscript still lacks a proper explanation regarding the assessment of the normality of data distribution. Moreover, the authors have not explained why they used some specific statistical tests. For example, why they used in some instances the Mann–Whitney U test and in others the Student’s t-test. Furthermore, the authors have included an inappropriate citation in the Statistical Analysis subsection: “Then we used the Mann–Whitney U test [21]”
In the Results section, especially in the Cohort Characteristics subsection, still numerous data presented in the text can also be found in the Tables.
